# A novel curation system to facilitate data integration across regional citizen science survey programs

Dana L. Campbell[1], Anne E. Thessen[2,3] and Leslie Ries[4]

[1] Division of Biological Sciences, School of STEM, University of Washington, Bothell, WA, USA
[2] The Ronin Institute for Independent Scholarship, Montclair, NJ, USA
[3] Center for Genome Research and Biocomputing, Oregon State University, Corvallis, OR, USA
[4] Department of Biology, Georgetown University, Washington, DC, USA

## ABSTRACT

Integrative modeling methods can now enable macrosystem-level understandings of biodiversity patterns, such as range changes resulting from shifts in climate or land use, by aggregating species-level data across multiple monitoring sources. This requires ensuring that taxon interpretations match up across different sources. While encouraging checklist standardization is certainly an option, coercing programs to change species lists they have used consistently for decades is rarely successful. Here we demonstrate a novel approach for tracking equivalent names and concepts, applied to a network of 10 regional programs that use the same protocols (so-called "Pollard walks") to monitor butterflies across America north of Mexico. Our system involves, for each monitoring program, associating the taxonomic authority (in this case one of three North American butterfly fauna treatments: *Pelham, 2014*; *North American Butterfly Association, Inc., 2016*; *Opler & Warren, 2003*) that shares the most similar overall taxonomic interpretation to the program's working species list. This allows us to define each term on each program's list in the context of the appropriate authority's species concept and curate the term alongside its authoritative concept. We then aligned the names representing equivalent taxonomic concepts among the three authorities. These stepping stones allow us to bridge a species concept from one program's species list to the name of the equivalent in any other program, through the intermediary scaffolding of aligned authoritative taxon concepts. Using a software tool we developed to access our curation system, a user can link equivalent species concepts between data collecting agencies with no specialized knowledge of taxonomic complexities.

Corresponding authors
Dana L. Campbell, danalc@uw.edu
Leslie Ries,
Leslie.Ries@georgetown.edu

## INTRODUCTION

There is a long history of piecing together multiple data sets to understand species distributions and patterns at large spatial and temporal extents, including the building of traditional range maps, range changes predicted by niche models (*Graham et al., 2004*) and the emerging field of macrosystems ecology that focuses on cross-scale dynamics

(*Heffernan et al., 2014*). Citizen science-powered monitoring programs now contribute prolific amounts of data towards these ends. Many programs carry out replicated surveys on permanent sites, using formal protocols similar to those in traditional scientific surveys (*Kelling et al., 2019*). These programs often operate on local scales to collect robust abundance, effort and absence data amenable for developing population indices across space and time, critical metrics for tracking factors such as population declines or shifts in phenology (*Pollock et al., 2002*; *Schwanghart, Beck & Kuhn, 2008*; *Pautasso & Weisberg, 2008*). For interpreting changes occurring across populations, there is great value in integrating data from monitoring programs operating at different spatial scales (*Downes et al., 2005*). This has motivated rapid development of integrative modeling methods to support large-scale analyses (*Pautasso & Weisberg, 2008*; *Levy et al., 2014*; *Zipkin & Saunders, 2018*; *Kéry, 2018*).

Despite newly-emerging analytical tools, actual implementations of integrative models are still rare. For example, in *Cooper, Shirk & Zuckerberg's (2014)* review of 171 macroscale studies of migratory birds, we note that while 72 of the studies (42%) used data from regional or continental-scale citizen scientist programs with protocols, 40% of those are based on a single source of data. Of the remaining studies, more than half used data from banding stations which benefit from centralized taxonomic standards followed by most participating programs. Published analyses from another highly monitored taxonomic group, the butterflies, also typically include only a single data source. Articles on US butterfly species, including studies examining the influence of environmental variables on entire butterfly communities (*Forister & Shapiro, 2003*; *Forister et al., 2010*; *Diamond et al., 2014*; *Cayton et al., 2015*; *Thorson et al., 2016*, *Wepprich et al., 2019*) all use data originating from a single program and thus did not require any data integration. In studies including data from more than one source (*Devictor et al., 2012*), the data from each source were analyzed separately (*Schmucki et al., 2016*). We could find only one published study (*Mills et al., 2017*) that performed an analysis integrating butterfly species concepts across multiple monitoring schemes.

Integrating data among multiple sources may be largely constrained by the challenge of consistent taxon matching. The clear trend in data collection is towards observation-based methods (such as citizen science monitoring programs) that do not generate vouchered specimen. *Amano, Lamming & Sutherland (2016)* documented that only 50% of all records in the Global Biodiversity Information Facility (GBIF) are associated with a vouchered specimen, and that number drops to 10% when looking just over the past decade. Furthermore, we found 98% of the records put into the GBIF-SCAN (Symbiota Collections of Arthropods Network, which is the main digitization portal for Lepidoptera occurrence records) in the last 10 years report human observation as basis of record (SCAN-GBIF analysis not presented). This means that in most cases names are the primary identifier for connecting comparable observations across diverse data sources. Integration of data stored in digital repositories such as the GBIF requires resolution of (1) names that contain typographical or formatting issues; (2) synonyms, that is, different names used for the same taxonomic concept; and, most complex, (3) names

that follow different authorities, that is, the same name used for different taxonomic concepts (*Boyle et al., 2013*).

For most species-level distribution research, standards for aggregating data consist of resolving deviant spellings, grammatical differences (e.g., hyphens and spaces), or synonymous names, but do not necessarily account for different taxon concepts circumscribed by a species name (*Kennedy, Kukla & Paterson, 2005*; *Giangrande, 2003*; *Pyle, 2004*, *Patterson et al., 2010*; *Mora et al., 2011*; *Patterson et al., 2016*; *Remsen, 2016*). Systematic and phylogenetic research continuously reclassifies the biological entities circumscribed by a name; thus unless different data sources explicitly reference the same (or compatible) authoritative treatments, it cannot be assumed that they monitor the same taxonomic concept even when they use the exact same name (*Ytow, Morse & Roberts, 2001*; *Pyle, 2004*; *Kennedy, Kukla & Paterson, 2005*; *Franz, Peet & Weakley, 2008*; *Franz & Peet, 2009*; *Boyle et al., 2013*; *Cui et al., 2016*; *Remsen, 2016*). This is problematic especially for unstable taxa, for which different taxonomic studies suggest reorganizations and disputed relationships (*Ytow, Morse & Roberts, 2001*).

*Franz et al. (2016)* illustrated the scope of this problem within the plant *Andropogon glomeratus-virginicus* "complex" (Poaceae). Careful examination of this species group revealed that multiple different sources of data collected over 126 years were attributed to a single species, but in fact the name under which these data were collected encompassed 12 different taxonomic concepts, because the sources collecting the data used different classifications of the taxa within this complex. *Franz et al. (2016)* found that names were reliable identifiers of taxonomic concepts in only 60% of the 12 pairwise alignments, indicating frequent mismatch between the taxonomic entities and nomenclature. Another study, based on the American Ornithological Union (AOU)'s North American checklist, shows that in the past 127 years 25% of North American bird names represent taxon concepts that were either split from or combined with another taxon, and of those names, almost half were then subsequently revised at least once (*Vaidya, Lepage & Guralnick, 2018*). An identical nomenclature used in gathering data before and after a revision may represent non-combinable records for non-identical taxonomic entities. We highlight an example of these issues for butterflies, the *Celastrina ladon* complex, in which, because of a history of multiple taxonomic interpretations, observations collected under the exact same name often do not actually represent the same species (Fig. 1). We also identify other problematic butterfly complexes encountered by North American monitoring programs.

Tools have been developed to crosswalk among naming systems, such as Avibase for birds (*Lepage, Vaidya & Guralnick, 2014*) and the Taxonomic Names Resolution Service (TNRS) for plants (*Boyle et al., 2013*), and standardized reference data sets are available to use for vertebrates (*Zermoglio, Guralnick & Wieczorek, 2016*).
The development of global taxonomic lists, such as the Integrated Taxonomic Information System (itis.gov) and Catalogue of Life (catalogueoflife.org) is another approach to standardize names across databases. More recently, the Global Names Architecture (GNA) was developed with the goal to find, index, and resolve taxonomic discrepancies among digitized taxonomic resources (*Pyle, 2016*). While these efforts have greatly increased our

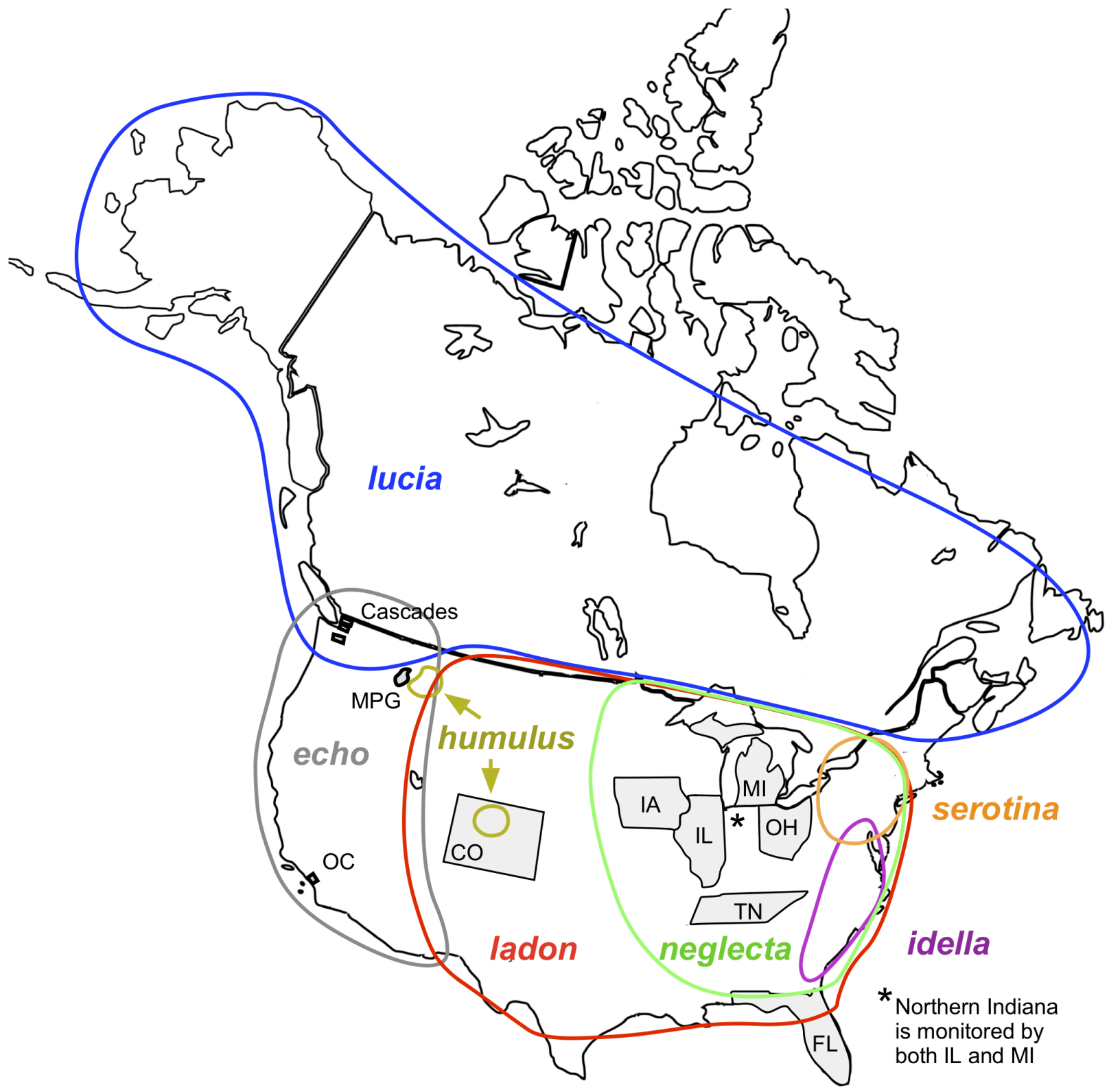

**Figure 1** **Map of North America showing locations of monitoring programs and *C. ladon* taxa (*Wright & Pavulaan, 1999*; *Pavulaan & Wright, 2005*).** For map credit see *Agyle (2013)*. Monitoring areas of the ten programs included in this article are shaded in gray. Program abbreviations are given in Table 4. Approximate geographical distributions of the seven taxa included in the *C. ladon* complex (colored circles) are shown as described in the original base lists (more recent taxonomic updates may conflict with these distributions). Note that *serotina* and *idella* do not occur within the monitoring areas of any of these programs.

ability to link knowledge across platforms focused on biodiversity data (*Patterson et al., 2016*), they have not solved all the problems related to data integration based on identity by name, even for the taxa for which the tools were created.

In this article, we present a curation-based solution for associating comparable taxonomic identities in the context of citizen science-powered monitoring, exemplified by North American butterfly monitoring programs. Throughout this article we use the term North American butterflies to mean those north of Mexico. A protocol for surveying butterflies was developed over 40 years ago specifically for volunteer monitors (*Pollard, 1977*). This was well before the current explosion of citizen science monitoring programs (see *Pocock et al., 2015* for development in Great Britain), and represents an early example of using structured protocols (i.e., where permanent routes are set up and monitored regularly using a set protocol, producing the most robust of survey data; *Kelling et al., 2019*). This protocol, referred to as the "Pollard walk" or "Pollard transect," involves monitors walking assigned routes at a steady pace and recording butterflies in a restricted observation window. The UK Butterfly Monitoring Scheme adopted the Pollard walk methodology in 1976 (*Schmucki et al., 2016*). Other European countries and

regions (usually states) in the US then followed throughout the 1980s, 1990s and 2000s (*Taron & Ries, 2015*; *Van Swaay et al., 2015*; *Schmucki et al., 2016*). In 2014, a web-enabled data entry platform called PollardBase (pollardbase.org) was launched to capture, manage, share, and archive North American butterfly monitoring data. PollardBase has since attracted the membership of most North American, Pollard-based programs (*Taron & Ries, 2015*).

For butterflies, most structured survey monitoring takes place on a regional basis, so any large-scale analyses will necessarily have to integrate data across programs (*Schmucki et al., 2016*). Any mismatches or confusion about names could lead to profound biases in the data, especially if a mismatch causes spurious zeros to appear (*Royle, Nichols & Kéry, 2005*; *Pautasso & Weisberg, 2008*). Inflated zeros strongly skew underlying data distributions and would fundamentally bias any analyses (*Kéry, 2018*). As large-scale analyses of both pattern and processes are becoming more common, especially through the emergence of the field of macrosystems ecology (*Heffernan et al., 2014*) and the advancement of integrated statistical models (*Zipkin & Saunders, 2018*), the need to make data interoperable without requiring each end-user to know the often convoluted taxonomic history of all the species in the regional community will become more and more vital (*Levy et al., 2014*; *Kelling et al., 2019*). Further, as integrated modeling schemes combine different types of distribution data, including counts, presence/absence, and presence-only data (*Kéry, 2018*; *Zipkin & Saunders, 2018*), the system we discuss here could potentially help ensure compatibility for curators of many different data types (e.g., iNaturalist, Butterfliesandmoths.org, eButterfly).

North American butterfly nomenclature is complex and controversial, and significant disagreements have spawned multiple differing lists of species names. As a case in point, estimates of the number of butterfly species in the United States and Canada range from 679 (*Scott, 1986*) to 780 (*Opler & Warren, 2003*) to 822 (*Pelham, 2014*). This is in stark contrast with the nomenclature of European butterflies and of North American birds. For birds, nomenclature is actively standardized by the AOU and the North American Classification Committee (NACC), and Avibase is an additional resource in resolving discrepancies (*Lepage, Vaidya & Guralnick, 2014*; *Chesser et al., 2016*; *Lepage, 2017*). Without any central taxonomic authority to reference, North American butterfly monitoring programs follow survey lists that reflect the idiosyncrasies, local expertise, and historical practices of their program. Thus naming discrepancies are much more common among butterfly datasets than bird programs and provide an excellent example for testing the effectiveness of our system.

To reduce confusion and ensure conformity of observations, it is common for monitoring programs to record data using their own unique and stable species checklist without regularly updating to newly published names or systematic reorganizations (*Ytow, Morse & Roberts, 2001*; *Taron & Ries, 2015*). Although willing to unite under a shared data management platform, PollardBase programs are disinclined to adopt a single standardized nomenclatural system and there is no way to compel them to do this. Even if it were possible, introducing a standardized nomenclature does nothing to resolve differences in legacy data that have accumulated over 40 years of monitoring in the US and

Canada. We thus developed our curation system to recognize and automatically match taxonomic entities across PollardBase programs, while allowing each program to retain their own unique checklist. All 21 programs that have joined PollardBase (including the 10 programs in this article) have agreed to adopt this system. Even in the unlikely case in which a group did not wish to cooperate, it would still be possible to code this group's list in a way that adheres to the underlying rules of our system for the purpose of combining data.

We built our system to resolve alternative names, spellings and taxonomic concepts between any two monitoring programs through the intermediary identification of each program with its underlying authoritative taxonomic treatment. Our system can also address the problem of recording cases where field identification is difficult and a program director chose to have volunteers record certain observations under a name that combines species. Additionally, our system is easily expandable, allowing any monitoring program to opt in at any point. By identifying compatible taxon concepts across programs, our system indicates which data can be combined, thus providing true data interoperability while "requir[ing] little specialized effort on the part of the end user," a major goal in the field of informatics (*Wilkinson et al., 2016*).

## Development of a novel name curation system

To evaluate the usefulness of our system, we show how it works in PollardBase and provide a proof-of-concept that the system can be used to automate the integration of data. Note that the PollardBase platform is designed so that volunteers collecting monitoring data for a particular program can choose only among the official species names included on that program's checklist (which were deliberately specified by the program director). Thus this study, based in the context of 10 PollardBase-registered (for current or future use) programs, allows us to quantify the challenges for data integration even in this ideal situation where the system controls for data entry-level introductions of typographical/spelling errors and for accidental use of an unofficial name (this is the most common source of name discrepancies in data collected outside of platforms like PollardBase). We targeted three specific objectives in developing and testing this new system:

- Presentation of a curation system based on an authoritative reference alignment for defining taxonomic concepts
- Quantification of the challenges for integrating data across 10 regional butterfly monitoring programs
- Development and testing of an automated integration tool to implement our curation system.

*Objective 1. Presentation of a curation system based on an authoritative reference alignment for defining taxonomic concepts.*

*Step 1: Identify authoritative lists of North American (US and Canada) butterflies and assign authority to each program's list.* We identified three research-based, taxonomic compendiums that consider the complete butterfly fauna of the United States and Canada:

**Table 1** Numbers of taxa recognized by each base list and ITIS.

| Base list | Subfamilies | Tribes | Subtribes | Genera | Subgenera | Species groups | Species | Subspecies | Total |
|---|---|---|---|---|---|---|---|---|---|
| Pelham | 23 | 46 | 37 | 247 | 38 | 36 | 820 | 1,570 | 2,817 |
| NABA | 24 | 0 | 0 | 221 | 0 | 0 | 725 | 112 | 1,082 |
| O&W | 22 | 2 | 0 | 231 | 0 | 0 | 784 | 2 | 1,041 |
| ITIS | 25 | 44 | 3 | 235 | 0 | 0 | 804 | 1,541 | 2,652 |

North American Butterfly Association (NABA) Checklist 2nd edition, version 2.3 (*North American Butterfly Association, Inc., 2016*) with associated commentary and interim reports (*Cassie et al., 2001*; *NABA Names Committee, 2015*, *2016*), henceforth referred to as NABA); *Opler & Warren (2003*; henceforth referred to as Opler and Warren or O&W*)*; and *Pelham (2014*; henceforth referred to as Pelham*)*. More information on these authorities is in Supplemental Article S1. We ascertained for the 10 PollardBase programs that we sample in this article (described in more detail below), that each uses a checklist similar to one of these authoritative treatments. In addition, we also include in our analyses the Integrated Taxonomic Information System (ITIS) species list (which we restricted to include just American butterfly species north of Mexico), because of its wider familiarity across the community of researchers and potential to serve as a link with other taxonomic databases. Table 1 shows the numbers of names each authority lists at different taxonomic levels.

PollardBase requests that each registered monitoring program provide its species checklist used for data collection in their locale, and, if known, the source of that list. The source could be a regional authority (e.g., a published regional field guide) or a national, research-based systematic authority. For programs that did not specifically base their list on NABA, Opler & Warren or Pelham, PollardBase administrators (which include all three authors of this article) determined which of these three base authorities it most closely resembled, and assigned that as the program's "base list." None of the 10 programs in this study declare ITIS as an authoritative basis for their own list.

*Step 2: Describing types of taxonomic deviations between base authority lists and in comparisons between programs and their base authority.* We manually aligned the three authoritative lists (NABA, Opler & Warren, Pelham) and the ITIS reference list to identify where nomenclatural differences lie. We used the notes published along with the authoritative lists to help interpret the taxon concepts represented by each list's taxon names and align equivalent concepts across the authorities (*Cassie et al., 2001*; *NABA Names Committee, 2015*, *2016*; *Opler & Warren, 2003*; J. Pelham, 2016, personal communication). Rarely, we also consulted further literature for clarification. Table S1 shows the alignment of all taxa that have one or more discrepancies among the four lists. This alignment acts as the backbone that ultimately allows us to associate taxonomic concepts between two programs whose checklists are based on different authorities (described in "Objective 3").

Next, we manually compared each program's species list with its specified authoritative base list to identify any differences. Programs were contacted to determine if identified deviations were intentional or if the program director might be willing to modify their list to match their authoritative base list. Note that this represented our only attempt to persuade programs to adopt a different nomenclature, as most programs were very clear before joining PollardBase that they would retain control over their species lists (L. Ries, 2012, personal communication with directors from all 10 North American butterfly monitoring programs included in this article). Program managers agreed to change the name to more fully adhere to their matching authority in about 50% of these cases. Reasons for not changing the name usually stemmed from local custom or the program director's (or local expert's) belief that the name on their list is the appropriate one for them (despite potential conflict with current taxonomic findings). This discussion with the program managers helped PollardBase administrators clarify the precise taxonomic concept represented by the name on the program list. In cases where we could not identify why the program's list deviated from their base list, we considered the program's discrepancy a discrete difference from its base and assumed the base authority's taxonomic concepts for all other non-divergent taxa. Occasionally ambiguities required us to ask program directors to interpret the implications of their deviation.

In all, we found six distinct types of deviations which we define, with examples, in Table 2. Some discrepancy types are relevant only between base lists, some only relevant for base list/program list comparisons, and some are relevant in both types of comparisons. We further quantify and characterize these discrepancies in Step 3, below. While many others have noted similar naming differences (e.g., *Vaidya, Lepage & Guralnick (2018)* discuss lumps and splits of taxa that would fall within and explain much of our "Species or subspecies conglomerates" category and issues of extralimital taxa that we categorize into "Unmatched taxa;" likewise *Ytow, Morse & Roberts (2001)* discuss differently circumscribed taxa), we have not seen as comprehensive a categorization of difference types among names as the one we lay out here.

*Step 3: Characterizing and quantifying taxonomic deviations among lists.*

*Pairwise comparisons between base lists*. Comparisons showed 603 species names that are identical (i.e., representing identical taxon concepts) across the three base authority lists and ITIS. This is 71% of the total number of 849 aligned taxon concepts. Differences in subfamily name were the most common type of deviations among base lists, but only because they were calculated on a per name basis for the entire classification hierarchy, so one change in subfamily name causes as many discrepancies as there are children taxa. Pairwise comparisons of base lists showed between 40 and 257 differences at the subfamily level. Subfamily names generally don't impact the issues of data integration because program lists are based on lower-level taxonomy, so hereafter we do not address issues of subfamily deviations but focus instead on pairwise differences at the genus and species level (with some subspecies differences causing species deviations). Table 3 shows the number of each deviation type that occurs between all possible pairwise comparisons of the base lists. ITIS and Opler & Warren lists are the most similar to each other overall, with a

**Table 2 Types of discrepancies observed in pairwise comparisons between authoritative base lists (NABA, O&W and Pelham) and between program checklists and their base authority.**

| Type of discrepancy (abbreviation) | Definition | Examples |
|---|---|---|
| Subfamily name deviation (F) | Lists use different subfamily names<br>*Definition relevant only for comparisons between base authority lists.* | O&W do not recognize subfamily Polyommatinae; all taxa designated as Polyommaninae by NABA, Pelham and ITIS are considered Lycaenidae by O&W. |
| Genus name deviation (G) | Lists use different genus names (usually species epithet is the same).<br>*Definition relevant for base list comparisons and base/program list comparisons.* | (1) ITIS, Pelham, O&W base lists distinguish genus *Zerene* from genus *Colias*, while NABA does not.<br>(2) Illinois list deviates from NABA base in using *Plebejus melissa* instead of *Lycaeides melissa*. |
| Species epithet deviation (S) | Lists use different specific epithets and/or subspecies usage, due to:<br>(1) differences in recognition of rank level for subspecies (subspecies promotion).<br>(2) errors of determination.<br>(3) different synonym is used.<br>(4) differences in subspecies use—One list extends to subspecies, the other does not.<br>Note: "Species" definition refers only to cases in which different species names are used purposefully; differences due to spelling are treated separately below.<br>*Definition relevant for base list comparisons and base/program list comparisons.* | (1) Subspecies promotion: NABA recognizes subspecies *Hesperia comma colorado*, while Pelham and O&W promote this to *H. colorado*.<br>Note: this means (a) different species names exist for the "*colorado*" taxon concept; and (b) the term "*H. comma*" represents more than one concept.<br>(2) Misdetermination: Hawaiian *Erionota torus* is listed as *E. thrax* by all base taxonomies except O&W, who note it was originally misdetermined as *E. thrax* by Linnaeus (1767).<br>(3) Different synonyms: ITIS, Pelham, O&W use *Piruna aea*, NABA uses *P. cingo*.<br>(4) Subspecies use deviations. In these cases, how/ whether names can be integrated across lists may depend on geography or concept: MPG lists *Speyeria cybele leto*, while all other monitoring groups list *S. cybele*. |
| Spelling/historical differences in scientific names (L) | Lists use alternate spellings.<br>*Definition relevant for base list comparisons and base/program list comparisons. As mentioned in text, program list spellings are sanctioned, not introduced by individual monitors.* | NABA list spells *Atrytonopsis edwardsi* whereas Pelham and O&W spell *A. edwardsii*. |
| Species or subspecies conglomerates (C) | One program list combines two or more species or subspecies together (that are separated in base lists).<br>*Definition relevant only for base/program comparisons.* | In every case encountered in our study, this divergence type occurred due to unreliability of distinguishing the taxa in the field: Illinois lumps *Colias eurytheme* and *C. philodice* (as *C. eurytheme/philodice*) although Illinois' base list (NABA) treats them as distinct species. |
| Unmatched Taxon (U) | Base or program list includes a species not on another base list, even under another name.<br>*Definition relevant for base list comparisons and base/program list comparisons.* | This deviation usually happens when a list includes a taxon that rarely/never occurs in North America: Pelham includes *Kisutam syllis* (a rare stray to southern Texas). |

**Note:**
Some comparisons showed more than one type of discrepancy.

total of 5% of names differing. Pelham and NABA, the most different lists, deviate at 23.5% of compared names.

As we show later, species type deviations include the most difficult discrepancies to resolve (although many are trivial). Species-level deviations among base lists affected up to 12.7% of all species. Genus and spelling discrepancies were rarer, up to 4.4% and 1.6% respectively, but these deviations are generally easily resolvable once detected because in all cases the alternative names align 1:1 with the taxonomic concept they represent.

**Table 3 Numbers of deviations by discrepancy type seen in pairwise comparisons of authority lists (including ITIS).** We calculate percentage for each discrepancy type as a function of the maximum number of species for each list pair (maximums shown in Table 1; for example, for Pelham/NABA comparison, species number is 820).

| | Genus deviation (G) | Species deviation (S) | Spelling (L) | Unmatched taxon (U) | Total deviations |
|---|---|---|---|---|---|
| Pelham/NABA | 35 (4.3%) | 104 (12.7%) | 12 (1.5%) | 42 (5.1%) | 193 (23.5%) |
| Pelham/O&W | 17 (2.1%) | 34 (4.1%) | 3 (0.4%) | 32 (3.9%) | 86 (10.5%) |
| O&W/NABA | 26 (3.2%) | 85 (10.8%) | 11 (1.4%) | 18 (2.3%) | 140 (17.9%) |
| ITIS/Pelham | 15 (1.8%) | 12 (1.5%) | 1 (0.1%) | 20 (2.4%) | 48 (5.9%) |
| ITIS/NABA | 29 (3.6%) | 94 (11.7%) | 13 (1.6%) | 25 (3.1%) | 161 (20.0%) |
| ITIS/O&W | 2 (4.4%) | 20 (2.5%) | 2 (0.2%) | 16 (2.0%) | 40 (5.0%) |

**Table 4 Deviations between each program list and its base list, by discrepancy type.** See Table 2 for definitions of discrepancy types.

| Program | Program abbreviation | Base list | Total taxa | Total deviations | Deviations per # taxa (%) | Genus name (G) | Species name (S) | Spelling (L) | Unmatched taxon (U) | Species |
|---|---|---|---|---|---|---|---|---|---|---|
| Cascades | CAS | Pelham | 151 | 8 | 5.3 | 4 | 2 | 1 | 1 | 0 |
| Colorado | CO | O&W | 245 | 10 | 4.1 | 0 | 6 | 2 | 2 | 0 |
| Florida | FL | O&W | 192 | 2 | 1 | 0 | 0 | 0 | 2 | 0 |
| Illinois | IL | NABA | 142 | 5 | 3.5 | 1 | 0 | 2 | 0 | 2 |
| Iowa | IA | NABA | 118 | 2 | 1.7 | 2 | 0 | 0 | 0 | 0 |
| Orange County (CA) | OC | O&W | 79 | 6 | 7.6 | 0 | 2 | 0 | 4 | 0 |
| Michigan | MI | NABA | 135 | 3 | 2.2 | 0 | 1 | 1 | 0 | 1 |
| MPG Ranch (MT) | MPG | Pelham | 116 | 4 | 3.5 | 3 | 0 | 2 | 0 | 4 |
| Ohio | OH | NABA | 165 | 7 | 4.2 | 1 | 1 | 4 | 1 | 0 |
| Tennessee | TN | NABA | 135 | 9 | 6.7 | 5 | 0 | 2 | 0 | 2 |

Deviations due to unmatched taxa are also relatively rare (up to 5.1%). We consulted the literature to assure that the missing taxon was actually excluded from the list and not included as a subspecies within another species' concept.

*Pairwise comparisons between programs and their base authority*. Table 4 shows the types of deviations found between the program lists and their base authorities. Florida's list was the most similar to its base, with 1% of names showing deviations (Table 4). Orange County (California) has the largest number of deviations for its list size (7.6% of names). As in base list comparisons, genus deviations and spelling differences between program and base lists were trivial to resolve. However, other deviation classes can be difficult to disentangle, especially if the person who compiled the list is not available to clarify a choice of nomenclature. Species name deviations and unmatched taxa represent the most difficult type of deviation to resolve because they often arise from fundamental disagreement about taxonomic concepts that are not always articulated in a published source. Species and species groups that have been revised multiple times can have a complex history of splitting/lumping disagreements (as demonstrated for birds in *Vaidya, Lepage & Guralnick (2018)*),

creating much confusion when the same species name is used by different programs to refer to a broader or narrower inclusion of subspecies and/or populations. Even if the nomenclature has been taxonomically resolved in the literature, programs retain their own species interpretations. This means that translating accurately from one program to another requires a thorough understanding of each program's interpretation of the nomenclature, complicating the process of establishing the match for each program pair.

*Objective 2. Quantification of the challenges for integrating data across 10 regional butterfly monitoring programs.*

One of our goals for this project was to quantify the challenges of taxonomic resolution problems across a sample of programs. For this implementation, we present just the simplest use case: that one of the 10 monitoring programs (the receiver) requests to receive data from one of the other monitoring programs (the donor) using the receiver's taxonomic interpretation to compare compatibility between the two programs. In reality, a more likely scenario is a third party requesting data from multiple programs. We do not consider this more complex case here, which requires the third party to provide their own taxonomic preferences for a final list, but consider this an important feature for future iterations of our tool.

The 10 regional butterfly programs included in this analysis cover a total of 489 distinct, comparable names, which we aligned manually (see Table S2). Of these, 82% either matched exactly across all programs in which they occurred (182 names), or were present in only one program list (208 names). In our alignment we took into account the taxon concept each of the 93 names (18%) that did not match exactly across programs. These discrepancies are categorized and summarized in Table 5, with more detail provided on the differences in Table S3. Most deviation instances were simple issues (category 1 in Table 5), in which programs names differed by simple spelling or synonym variants, but aligned 1:1 with taxon concept. More difficult were subspecies issues (category 2 in Table 5), where some programs listed names that did not align 1:1 across comparisons. The third category of deviations in Table 5 shows the most difficult discrepancies, including conglomerates and complexes, which present much more difficult alignment issues.

Our alignment of program names guided our development of a set of species concept matrices that specify the comparable taxonomic unit for every possible pairwise combination in which one of the 10 programs (receiver) requests data from another program (donor), based on the receiver's taxonomic perspective. We use our set of matrices to quantify the challenges of integration and also as a validation check for our translation tool (Objective 3).

As an illustration of our process to determine comparable taxonomic units across programs, which starts by identifying taxa that do not exactly align and then translating each case into a taxon concept matrix, we discuss here one of our most difficult taxonomic cases, the *Celastrina ladon* complex. This complex is composed of seven taxa (approximate ranges shown in Fig. 1), which are treated differently by each of the authoritative base lists. The *C. ladon* complex is problematic partly because the taxonomic entities can be difficult to distinguish morphologically. Pelham and Opler & Warren

**Table 5 Summary of discrepancies across monitoring programs.** For more detail, see Table S3. Note: issues here are those that pertain only to our 10 program lists. Authority list comparisons have taxonomic conflicts between taxa that are not sampled by these program lists.

| Discrepancy types | Number of incidences | Number of names used | Number of concepts | Notes |
|---|---|---|---|---|
| 1. Simple issues (names align 1:1 with taxon concept) | | | | |
| Genus deviations: different genus names used | 8 | 16 | 8 | Combinable. |
| Species deviations: alternate synonym or spelling used | 8 | 19 | 8 | Combinable. For one taxon (*Phyciodes selenis*), programs used four name deviations that we consider (combinable) synonyms, however users should be aware that other researchers may recognize taxon concept distinctions among these terms. |
| 2. Subspecies issues | | | | |
| Some programs specify taxon to subspecies level (names align 1:1 with concepts at all comparisons) | 10 | 20 | 20 | Combinable at species- and/or subspecies-level comparisons when appropriate subspecies data is parsed at both programs. User should ensure level of analysis is clear. |
| Some programs list more than one subspecies (subspecies names do not align 1:1 with species-level concept) | 2 | 6 | 6 | Combinable at species- and/or appropriate subspecies-level comparisons.<br>Both subspecies must be included in species-level comparisons with other programs. User combine with care. |
| 3. Taxonomic issues: identification, alternate species names, taxonomic revisions, subspecies promotion interpretations (names do not align 1:1 with taxonomic concept) | | | | |
| Conglomerates (species-level ID difficult to distinguish in field so some programs collect combined data) | 2 | 6 | 6 | For some program combinations, data must be analyzed at levels higher than species concept.<br>A third taxon conglomerate occurs in *C. ladon* species complex; numbers of names and concepts for this conglomerate are included in complexes category below. |
| Complexes (different taxon concept interpretations) | 7 | 19 | 18 | Includes issues of subspecies promotion, taxonomic revisions and different interpretations of species concepts. Tool interprets compatibility based on the taxonomic interpretation of program receiving data and geographic range when appropriate. |
| *Celastrina ladon* species complex | 1 | 7 | 8 | Within the *C. ladon* species complex some programs also conglomerate taxon collection for *ladon* and *neglecta*. |

elevate all taxa to full species status (although Opler & Warren does not treat *serotina*), whereas NABA considers all as subtaxa within *C. ladon*. In total, our programs monitor five of these taxa (Table 6). Because of conflicting identities, determination of data compatibility between two programs requires careful consideration of the taxonomic concepts used by each program, even when the recorded names are identical.

For instance, as shown in Figs. 1 and 2, both *ladon* and *neglecta* occur in the area monitored by the Iowa program. Iowa, following NABA's taxonomic interpretation, considers these taxa as subspecies, and collects observations for both together under the species name "*C. ladon*." Conversely, the Florida program, basing its perspective on Opler & Warren's taxonomy, takes observations of *C. ladon* and *C. neglecta* separately, as distinct species. Thus, despite sharing the same name, Iowa's "*C. ladon*" observations are not compatible with "*C. ladon*" observations from Florida. Iowa might consider data compatible only when Florida's data from both *C. ladon* and *C. neglecta* are included in comparison with Iowa *C. ladon* data, however a researcher wanting to integrate data

**Table 6 The taxonomic entities from the *C. ladon* complex occurring in each program range, and nomenclature used by each program.** (D) indicates that the program's interpretation deviates from that of its declared authoritative base. Boxed names indicate cases where a program uses a single name to circumscribe a butterfly population that other programs split into two separate names. E.g. Illinois and Iowa both lump *ladon* and *neglecta* together under one name (*C. ladon/neglecta* and *C. ladon*, respectively) whereas Ohio splits this concept into two names (*C. ladon* and *C. neglecta*). Dorsal (left wing) and ventral (right wing) images show wing variation among taxa (although much variation also exists within each taxon, a source of the taxonomic confusion). No dorsal image available for *humulus*. Photo credits: *C. ladon*, *C. neglecta*, *C. lucia*, *Schmidt & Layberry (2016)*; *C. echo*, CBG Photography Group, Centre for Biodiversity Genomics, CreativeCommons BY-NC-SA; *C. humulus*, Jeffrey Pippen.

| Program | Base authority | Taxonomic entity and nomenclature used by each program | | | | |
|---|---|---|---|---|---|---|
| | | ***ladon*** | ***neglecta*** | ***lucia*** | ***echo*** | ***humulus*** |
| | | 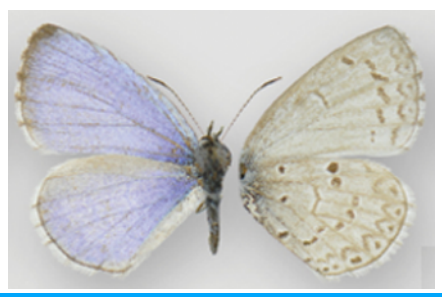 | 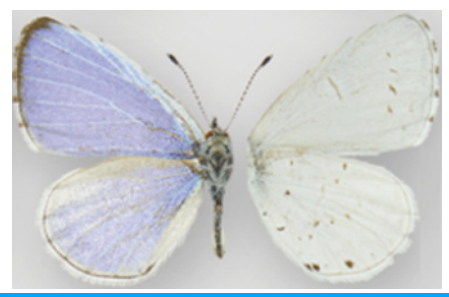 | 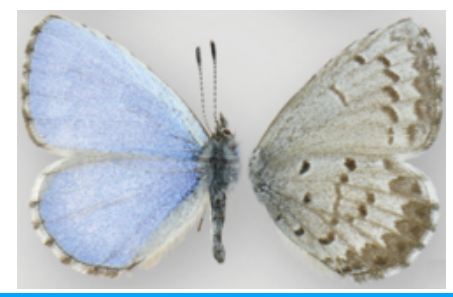 | 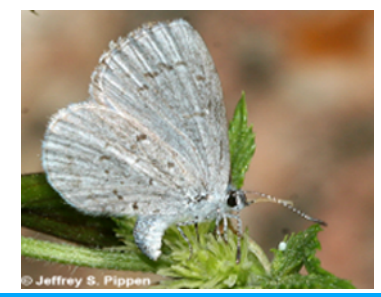 | 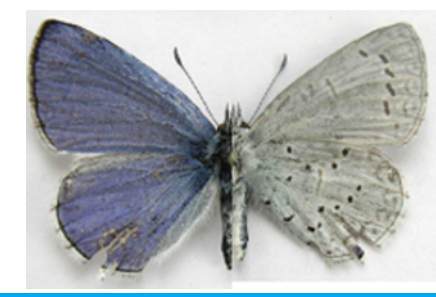 |
| Illinois | NABA (D) | *C. ladon/neglecta* | | | | |
| Iowa | NABA | *C. ladon* | | | | |
| Michigan | NABA (D) | *C. ladon/neglecta* | | | | |
| Ohio | NABA (D) | *C. ladon* | *C. neglecta* | | | |
| Tennessee | NABA (D) | *C. ladon* | | | | |
| Florida | O&W | *C. ladon* | *C. neglecta* | | | |
| Cascades | Pelham | | | *C. lucia* | *C. echo* | |
| MPG Ranch, MT | Pelham | | | | *C. echo* | |
| Colorado | O&W | *C. ladon* | | | | *C. humulus* |
| Orange Co, CA | O&W (D) | | | | *C. ladon echo* | |

could easily overlook this compatibility complication. From Florida's perspective, Iowa's data are not compatible at the species level for either *C. ladon* or *C. neglecta* individually. Similarly, the Cascades program, following Pelham's taxonomy, collects distinct observations for *C. lucia* and *C. echo* as separate species. Thus, from Cascades' taxonomic interpretation, there are no data from the NABA-based Iowa program that is compatible with either *C. lucia* or *C. echo*. On the other hand, Iowa interprets combination as valid as long as both *echo* and *lucia* data from Cascades are included with Iowa's *C. ladon* data. These examples show how the taxonomic interpretation of the monitoring programs fundamentally affect compatibility of the data they collect; data may be compatible between programs when one of the programs is the recipient but not when the other program is; that is, requests may not have a symmetrical solution when flipped the other direction. Furthermore, in the case of the *C. ladon* complex, half the monitoring programs deviate from the interpretation of their own base authority (see Table 6), emphasizing the potential complexity of deciphering compatibility between program observations.

We built a species concept integration matrix for the *C. ladon* complex indicating the compatibility of data collected by each receiver program with data collected by any of

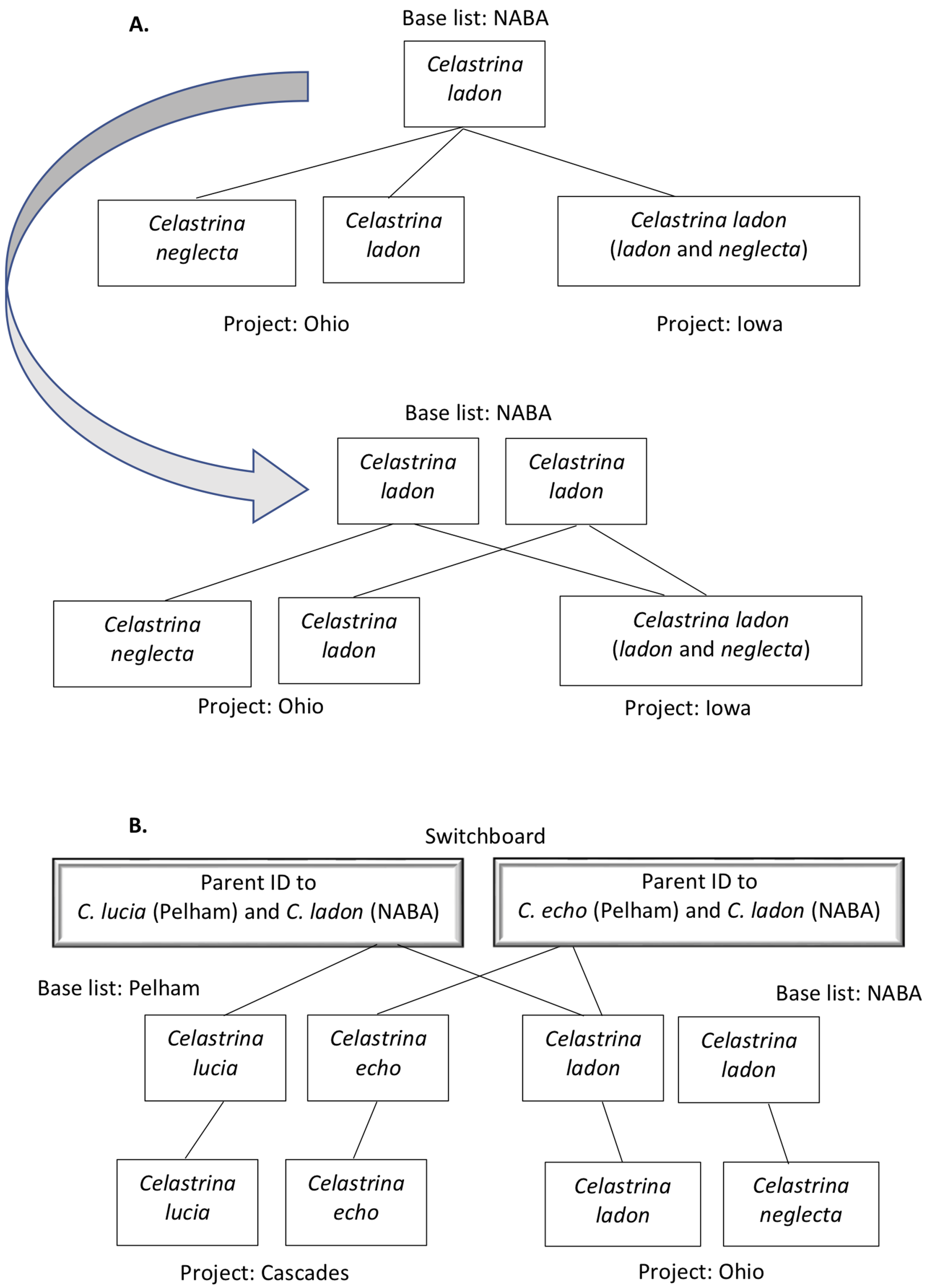

**Figure 2 Examples of taxa translation across programs in the tool's name-mapping data file.** In some cases, a program list deviates from its base list such that multiple taxa on the program list match to a single species on the corresponding base. For example, Ohio's list includes *C. ladon* and *C. neglecta*. (A) Top panel illustrates that if we were to directly translate at the species level from Ohio's taxa to its (NABA) base equivalent, we would, in the process, eliminate Ohio's differentiation of *C. ladon* and *C. neglecta*, since NABA considers these subspecies of *C. ladon*. To retain granularity and manage relationships that are not differentiated by name string, we duplicate the name in the base list, allowing mapping of each

**Figure 2** (continued)
program concept to a distinct base list holder (second panel, end of gray arrow). (B) When two programs have different base lists, we translate between them by introducing a "switchboard" that manages relationships and granularity between taxa in the base lists. Our algorithm (demonstrated in Fig. 5) uses this structure to find equivalent names between programs.

the nine potential donor programs (rows and columns, respectively, in Table 7). When the donor program uses the same epithet *and* taxonomic concept as does the receiver, the outcome is coded in the matrix as a Perfect match (PM). Where programs do not have an exact match, the receiver's request results in one of the following outcomes:

- No match (Zero). The taxonomic unit does not occur at the donor program. This means that receivers can assume that abundances are always zero for that species throughout the donor program's range.
- Compatible match (CM). Compatible matches connect observations that represent the same taxonomic concept as specified by the receiver, but symbolized by the donor with a different nomenclature. This could be due to the use of a different genus or species epithet, the adoption of alternate spelling, or the differential recognition of species/subspecies taxonomic levels.
- Multiple match (MM). When there are differences in lumping and splitting taxonomic entities, one program may pool together, under a single name, a concept that is separated into two species by another program. If the program that lumps under one name requests data from a program that splits the species into two, the tool will return both species, which we designate here a "Multiple match." In the reverse case, where a program that separates the taxa into two names requests a match for one of the names from a program that lumps the taxa together, we flag the response so that the user knows that donor data are only combinable when data from both receiver taxa are included (and thus may not represent a species-level entity from the receiver's perspective). These cases are indicated in Table 7 as CM+.

Table S4 lists the 39 species concept integration matrices for all other taxa (18% of taxon entities) where there are taxonomic names and/or concepts mismatches between programs.

*Objective 3. Development and testing of a novel integration tool.*

Using our alignment of program names in combination with integration matrices, we built a tool for users to answer, on a case by case basis, whether it is appropriate to integrate data associated with any particular taxon listed on any monitoring program and data gathered by any other monitoring program. The tool is currently available through GitHub (https://github.com/diatomsRcool/butterfly) and mybinder (http://mybinder.org/repo/diatomsRcool/butterfly). This proof-of-concept tool is currently a stand-alone tool, but one that we intend to make fully operable for automating data integration within the PollardBase citizen science butterfly monitoring project infrastructure and can also be implemented on other appropriate portals, such as butterfliesandmoths.org. Thus, the user

**Table 7 Data integration matrix for the *Celastrina ladon* complex showing matches between each pairwise combination of monitoring programs.** In total, our ten monitoring programs use five taxonomic names (*ladon*, *neglecta*, *echo*, *lucia*, *humulus*) to represent multiple taxon concepts in the *C. ladon* complex. No program contains more than two taxa within its range. Each cell in the matrix indicates the outcome of a request for equivalent data from the "Receiver program" (R) in rows, to the "Donor program" (D), in columns. Where there is no match in the donor program, the cell is marked "Zero." Types of match (CM, PM, MM) are defined in text. A plus symbol (+) next to the match type indicates that the receiver program must include data from more than one of its surveyed taxa for compatibility with the donor's match, thus the receiver may consider the donor's content as not combinable at their species-level interpretation. A pair of programs may have compatibility in one direction but not in the other (that is, if donor and receiver roles are reversed). Gray cells indicate situations where the match is not symmetrical in both directions of request. Comparisons between programs that share NABA as their base list (top left quadrant, boarded with thick lines) have less conflict than comparisons between programs whereby one program has a NABA base and the other program has Pelham or O&W base list.

| | ***Celastrina ladon* complex taxa occurring at each program (Base authority)** | **Donor**<br>**IL (NABA)** ***C. ladon/ neglecta*** | **IA (NABA)** ***C. ladon*** | **MI (NABA)** ***C. ladon/ neglecta*** | **OH (NABA)** ***C. ladon C. neglecta*** | **TN (NABA)** ***C. ladon/ neglecta*** | **FL (O&W)** ***C. ladon C. neglecta*** | **CAS (Pelham)** ***C. lucia C. echo*** | **MPG (Pelham)** ***C. echo*** | **CO (O&W)** ***C. ladon C. humulus*** | **OC (O&W)** ***C. ladon echo*** |
|---|---|---|---|---|---|---|---|---|---|---|---|
| **Receiver** | **IL (NABA)** *C. ladon/neglecta* | X | CM | PM | MM | PM | MM | MM | CM | MM | CM |
| | **IA (NABA)** *C. ladon* | CM | X | CM | MM | CM | MM | MM | CM | MM | CM |
| | **MI (NABA)** *C. ladon/neglecta* | PM | CM | X | MM | PM | MM | MM | CM | MM | CM |
| | **OH (NABA)** *C. ladon* | CM+ | CM+ | CM+ | X | CM | CM | MM | CM | MM | CM |
| | *C. neglecta* | CM+ | CM+ | CM+ | | CM+ | PM | Zero | Zero | Zero | Zero |
| | **TN (NABA)** *C. ladon/neglecta* | PM | CM | PM | MM | X | MM | MM | CM | MM | CM |
| | **FL (O&W)** *C. ladon* | CM+ | CM+ | CM+ | CM | CM+ | X | Zero | Zero | PM | Zero |
| | *C. neglecta* | CM+ | CM+ | CM+ | PM | CM+ | | Zero | Zero | Zero | Zero |
| | **CAS (Pelham)** *C. lucia* | Zero | Zero | Zero | Zero | Zero | Zero | X | Zero | Zero | Zero |
| | *C. echo* | Zero | Zero | Zero | Zero | Zero | Zero | | PM | Zero | CM |
| | **MPG (Pelham)** *C. echo* | Zero | Zero | Zero | Zero | Zero | Zero | PM | X | Zero | CM |
| | **CO (O&W)** *C. ladon* | Zero | Zero | Zero | CM | Zero | PM | Zero | Zero | X | Zero |
| | *C. humulus* | Zero | Zero | Zero | Zero | Zero | Zero | Zero | Zero | | Zero |
| | **OC (O&W)** *C. ladon echo* | Zero | Zero | Zero | Zero | Zero | CM | MM | CM | CM | X |

interface is, as yet, underdeveloped. The tool consists of two parts: (1) a file that contains the mapping between each program list and their base list and from the base lists to a master switchboard list (see Fig. 2); and (2) an algorithm that takes user input and uses the mapping file to return the requested results. The following text describes in detail the mapping and the algorithm.

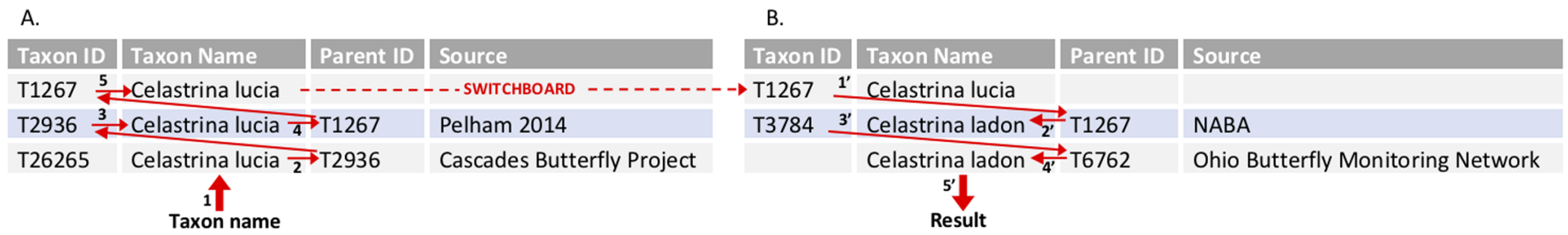

**Figure 3 Tool algorithm for translating between two program lists through the switchboard—example query.** This query uses the mapping file described in Fig. 2B. The user has entered three pieces of information: a taxon name (*C. lucia*); the "Recipient program" (Cascades Butterfly Project); and the "Donor program" (Ohio Butterfly Monitoring Network). (A) The algorithm first works its way up the hierarchy to the switchboard, starting with the input taxon name *C. lucia* (1) and "Recipient" program (Cascades Butterfly Project) to find the appropriate data row in the mapping file. The parent identifier (2) for *C. lucia* at Cascades (T2936) is used to find the row for the corresponding taxon in the Pelham base list (3). The parent identifier for the base list (4; T1267) acts as the taxon ID at the switchboard (5, and dashed red line). (B) The algorithm searches for rows that have the switchboard taxon identifier as the parent identifier (1′). These results are filtered, because the user identified a "Donor program," (Ohio) which has a defined base list (i.e., the algorithm knows to look only for a row where NABA is the source). Once the appropriate base list taxa have been found, its taxon identifier (2′; T1267) is used to find its child in the (Ohio) project list (3′; T3784). The name associated with the appropriate parent identifier (4′ T6762) is returned as the result (5′; *Celastrina ladon*).

### *The mapping*

We prepared a data file that links names in each program's checklist to the corresponding name in the program's base list. This file also links corresponding base list names across authorities in the master switchboard list. These mappings are codified using parent/child relationships conforming to the Darwin Core biodiversity standard (*Wieczorek et al., 2012*). This initial linking was done manually as described in Steps 2 and 3 in Objective 1 above. Figure 2 shows two examples for how these links connect names between two program lists through their associated base lists. Figure 2A shows an example of how we adjusted lists in the mapping file (lower panel, at head of arrow shows we made duplicate *C. ladon* entries), so that they are not exactly true to the original lists (upper panel). This duplication is necessary to account for multiple taxon concept interpretations. Figure 2B shows the mapping of taxa between program lists that have different base lists, using a master switchboard list to (1) disambiguate and manage the multiple concept interpretations of the same name between base lists and (2) allow us to protect the granularity of a program list that may use a less specific base list.

As is standard for Darwin Core representations of biodiversity classifications, every name on each program and base list included in our mapping file is assigned a unique Taxon ID, Parent ID, and a Source (*Wieczorek et al., 2012*). Each name in a program list has a parent in the base list. Each name in the base list has a parent in the switchboard. The Taxon ID and the Parent ID are alphanumeric strings. The Source will always be either a program list or a base list reference. It is this chain of Taxon ID and Parent ID that the algorithm uses to travel from a Start Program to the switchboard and back down to the End Program.

### *The algorithm*

The algorithm takes three pieces of information as input from the user: the recipient program, the taxon of interest, and the donor program. It then uses the mapping file to

return the requested result—the equivalent taxon of interest from the donor program. The algorithm searches for the taxon of interest in the recipient program using string matching and then uses the parent identifier to move up the parent/child hierarchy to the switchboard (e.g., Fig. 3A). After reaching the switchboard, the algorithm starts traveling back down the parent/child hierarchy by searching for taxa with a specified parent identifier until it finds the appropriate taxon or taxa in the donor program (Fig. 3B). If no compatible taxa can be found, the system returns: "There is no match in this list (species presumed absent)." This is the case for requests between programs that show "Zero" as the outcome in the concept integration matrix.

### *Tool outcomes*

In the cases where the matching is too complicated to capture in an algorithm, the tool returns a flag along with the outcome to explain the confusion for that particular pairwise comparison or to suggest that the outcome requires the user to make an individual determination on whether the two taxon concepts are indeed compatible before combining data. For technical reasons, the tool must give symmetrical returns for both directions (i.e., when donor and receiver roles are exchanged) in a pairwise comparisons of programs. In cases where the species matches are not symmetrical in both directions (gray cells in Table 7), the tool returns a flag. Supplemental Article S2 details these two types of flagged outcomes. Note that the tool flags results of difficulties in cases where there is substantial disagreement, even though in our equivalence matrix the comparison might be designated as Compatible or Multiple match. The *C. ladon* complex is complicated by a broad diversity of concept interpretations among base lists and program lists, and every pairwise comparison except two (requests between OC and MPG) results in a flag accompanying the taxon outcome (Table 7).

### *Testing the tool*

We carried out comprehensive testing on pairwise comparisons across all taxa in all programs, shown in Table S5. Each comparison was listed with its expected results. Tests were submitted either in bulk on the command line or singly in a Jupyter notebook. A test was successful if the tool gave a result that matched what was expected based on our pre-constructed compatibility matrices. Refinement of the tool occurred iteratively until 100% of comparison tests were successful.

## DISCUSSION

We have developed a system for the curation of names and associated taxonomic concepts for programs that actively collect data over long time periods. This allows data to be easily integrated between any monitoring programs that adopt this system, even when they use different or even unique nomenclatures. While there are other available systems to resolve taxonomic name conflicts of existing data, they are all retrospective and often rely on assumptions about the collector of the data, who may or may not be available to articulate their concepts when they affixed a name to a data point. Our system is proactive in that it coordinates programs that are continually collecting data by registering their own specified taxonomic names and concepts in relationship to a standardized checklist.

By ensuring that PollardBase program directors themselves retain the control of associating taxonomic names between their checklist and base list, we eliminated any disinclination or barriers programs might have in signing on to use our system, and in fact rather than resenting our intrusion, all programs appreciated our help in curating their species lists.

Combined with PollardBase, which prevents the accidental creation of incorrect name text strings, this system creates a novel taxonomic curation and integration system that maps all, including legacy, concepts between programs while making no demands on the involved monitoring programs and requiring little taxonomic knowledge on the part of the end user. Our testing shows that our tool can be used to implement our curation system. Using 10 North American monitoring program checklists and three taxonomic authority lists, our tool accurately recovered comparable taxonomic units between monitoring programs when programs identify taxa at the same taxonomic resolution. When species are identified at different taxonomic levels, our tool flags the data as non-combinable and identify where there may be problems of spurious zeros. Developing a taxonomic backbone that could ensure 100% correct integration under every situation is extraordinarily complex. As currently designed our tool does not provide a solution for mismatches, but instead leaves it up to the individual researcher to resolve them. Other systems have been designed to deal with this problem (see *Ellingsen et al., 2017* for one example) and there is room for improving specific integration guidance for these more complex situations in future iterations of this tool.

A key advantage of our system is its scalability. Adding a new monitoring program only requires the program to declare one of the taxonomic authorities as its base, and to register any deviations from it. Once that is done, the tool automatically determines name compatibility between the new and pre-existing programs as described above. Similarly, if a new authority list is published that the community begins to use, we can add it as a new base list simply by encoding its relationships via our switchboard. While the potential for much more complicated relationships between concepts increases with additional programs, the structure of the mapping file makes future interoperability tractable.

Currently in early stages of development, our tool is a beginning for implementing our curation framework for associating combinable data. Important improvements would include enabling the tool more generally and make it more easily operable by assimilating it with a more developed interface into the PollardBase and other appropriate platforms; expanding its flexibility by allowing users more customizable taxonomic interpretations; extending taxonomic interoperability to include the perspective of independent third parties interested in the data from these programs; and developing the means for automated updating of taxonomic concepts as they arise in the literature. As case in point, we note the recent revision of the *Anthocharis* species complex (*Stout, 2018*) is not reflected in any of our base authorities (although Pelham's most recent (2019) version of his species catalog does include Stout's revision); we updated by flagging the user to consider potential discrepancies.

Although developed for and tested on the PollardBase network of butterfly monitoring programs, there is nothing about this system that is particular to butterflies. The utility of our system could be implemented to help solve the significant stumbling block of inaccurate taxonomic integration across structured monitoring programs focused on any taxonomic group for which: (1) a taxonomic authority can be identified as the basis for each program's species checklist; (2) any differences among these authorities can be mapped to build an infrastructure to translate between programs; (3) programs can state any deviations between their checklist and their adopted base list. Regional structured surveying of taxa is becoming increasingly popular (*Kelling et al., 2019*), and taxa such as odonates and amphibians are particularly amenable to surveys using Pollard-like protocols. Indeed, Pollardbase began accepting odonate and amphibian survey program participants in 2018 and 2019 respectively.

Citizen science monitoring observations are one of the only sources of large-scale spatiotemporally-replicated biodiversity data available to answer our most important global change questions. Increasingly complex and powerful models are available to integrate data among regional monitoring programs but accurate results depend on consistent nomenclature to prevent systemic mismatches. Our system is an innovative approach to identify and resolve regionally employed taxon concepts by curating them alongside published authoritative treatments and will guide researchers in tracking North American butterfly populations, while preventing erroneous ecological conclusions drawn from incorrectly associated taxa.

## ACKNOWLEDGEMENTS

The authors would like to thank the directors of each of the participating butterfly monitoring programs. Doug Taron especially helped us to develop a system that would be tractable to implement. Program directors at the time we implemented these programs include Doug Taron (Illinois), Jerome Wiedmann (Ohio), Ashley Cole-Wick (Michigan), Sarah Garret (Colorado), Jaret Daniels (Florida), Nathan Brockman (Iowa), Jutta Burger (Orange County), Regina Rochefort (Cascades), Jeff Pippen (MPG Ranch) and Steve McGaffin (Tennessee). We thank Rob Guralnick and Jeff Pippen for helpful feedback on the project and Jeff Pippen for helping to implement the curation system for new users of PollardBase.

### Funding

Funding for this project was provided by NSF (DBI-1147049). The funders had no role in study design, data collection and analysis, decision to publish, or preparation of the manuscript.

### Grant Disclosures

The following grant information was disclosed by the authors:
NSF: DBI-1147049.

## Competing Interests

Leslie Ries is an Academic Editor for PeerJ.

## Author Contributions

- Dana L. Campbell conceived and designed the experiments, performed the experiments, analyzed the data, prepared figures and/or tables, authored or reviewed drafts of the paper, and approved the final draft.
- Anne E. Thessen conceived and designed the experiments, performed the experiments, analyzed the data, prepared figures and/or tables, authored or reviewed drafts of the paper, and approved the final draft.
- Leslie Ries conceived and designed the experiments, performed the experiments, analyzed the data, prepared figures and/or tables, authored or reviewed drafts of the paper, and approved the final draft.

## Data Availability

Our tool and its code is available through GitHub (https://github.com/diatomsRcool/butterfly) and mybinder (http://mybinder.org/repo/diatomsRcool/butterfly).

## Supplemental Information

Supplemental information for this article can be found online at http://dx.doi.org/10.7717/peerj.9219#supplemental-information.

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
