# Peer review of "A novel curation system to facilitate data integration across regional citizen science survey programs"

_PeerJ, doi:10.7717/peerj.9219_

## Round 0.1 · original submission · Major Revisions

All reviewers agree that you address an important problem for programs, citizen science here but more generally any monitoring programs struggling with various taxonomic databases and resolutions over time. I have myself been confronted to this in a marine monitoring program - temporal changes of benthos communities were in analyses of raw data linked to different taxonomic decisions associated to different laboratories (Ellingsen, K. E., N. G. Yoccoz, T. Tveraa, J. E. Hewitt, and S. F. Thrush. 2017. Long-term environmental monitoring for assessment of change: measurement inconsistencies over time and potential solutions. Environmental Monitoring and Assessment 189:595). Reviewers have made constructive comments that will help you revise the manuscript.

Reviewer 1 ·

Basic reporting

See comments below

Experimental design

See comments below

Validity of the findings

See comments below

Additional comments

The concept of this manuscript is interesting and worthwhile. However, I would like to see less "it's difficult" and more consideration of standardizing the taxonomy of individual programs and assessment of other sources of bias from individual or integrated programs, especially those that are not implemented primarily by experts.

The title is quite long and indirect. Try something such as “Integration of taxonomic classifications across regional citizen science programs.”

Please double-space manuscripts even when not required by a given journal. Double-spaced manuscripts are much easier to read.

The abstract is rather long. Aim for no more than 300 words.

The manuscript as a whole is unnecessarily discursive. For example, no need to explain that Linnaeus developed binomial nomenclature (line 65).

Don’t conflate the concepts of extent and resolution. E.g., at line 60, “large scale” is accurate if the intent is large area and coarse resolution. If the intent is large area, then it’s “large extent.” See also line 104 and elsewhere.

line 63 and elsewhere. Reserve “we” for the authors.

The manuscript is missing an opportunity to underscore how critical voucher specimens are for accurate identifications, especially when observers do not have formal training in the taxonomic group and when population viability almost certainly will not be affected by taking vouchers, as is the case for virtually all butterfly species.

line 79. Change to “eBird; however,”

line 81. “scientific” is a more inclusive and honest descriptor than “academic”

line 89. Integrative modeling is not really new, and Zipkin and Saunders 2018 don’t claim that it’s new. Zipkin and Saunders wrote about integrated population models, and noted that variations of these models have been used since the early 1980s.

line 90. It’s not clear to me why false negatives are more problematic, at least with respect to scientific inferences, in abundance models than in occupancy models. Moreover, many formal surveys are not presence-only, but presence-absence.

line 137. Here and elsewhere, “species” is more accurate than “biodiversity.” Biodiversity includes all levels of life, e.g., genes to biomes, and encompasses structure, composition, and function

line 148. For a general audience, might be good to note that these are plants.

lines 148 and 155. Fair enough, but in every taxonomic group, there are taxa that simply are problematic taxonomically. In the United States, not only some Celastrina groups but many Speyeria and Papilio groups are good examples. Moreover, in every taxonomic group, there are taxa that are extremely difficult to identify. For instance, some butterflies can be identified definitively only by examination of genitalia.

line 154. Invertebrates, or just butterflies?

line 160. North America extends to the southern border of Panama. Is that what you mean, or do you mean “United States and Canada,” or even “United States”?

line 163. This seems like a straw man to me. Virtually all serious lepidopterists, including Warren, are now deferring to Pelham. Scott’s Butterflies of North America is an excellent reference in many ways, but it is not currently regarded as a taxonomic authority. It’s unfortunate if some program administrators and observers are unaware of that, but I think a strong case can be made for increasing awareness.

line 167. Now AOS (American Ornithological Society)

lines 159 and 173. I’m having some difficulty identifying the objective of the manuscript. Is it to standardize butterfly nomenclature? Is it to promote a particular method? One might disagree that retaining distinct local nomenclature is useful if the objective is to obtain rigorous data. At line 173, I would prefer “for testing the effectiveness.”

line 176. Habitat is a species-specific concept that should not be confused with vegetation type or land-cover type (either of which would be more accurate here)

lines 178-181. This is a cliché and not true. Butterflies indeed are highly responsive to environmental variability. As a result, their responses to any given environmental change are quite difficult to predict or interpret. Additionally, there is very little evidence that responses of butterflies to a given change can be used as a surrogate for responses of other taxonomic groups.

line 182. Keep in mind that there are two major objectives of citizen science. One is to engage the public, and butterflies probably are good for that. The other is to conduct rigorous science, and whether data collected by non-scientists are reliable for understanding butterfly ecology is unclear. This especially is the case when observers or programs are reluctant to take voucher specimens.

line 185. This might be phrased differently. “Fluctuations” may not be the best word. The abundance of a given species changes throughout the season as a result of, for example, staggered emergence, protandry, and multiple broods. Abundance trajectories may be difficult to project, but one knows that abundance will vary. And, of course, different species have different trajectories.

line 205. Legacy data can be reclassified.

line 221. Better to aim to test usefulness in a scientific manuscript than to aim to demonstrate usefulness.

line 234. Yes, it’s difficult. Demonstration of difficulty as an actual objective seems gratuitous.

line 242. Are these two paragraphs basically saying that four lists were considered authoritative? Simplify.

line 246. Again, Pelham generally is considered to be the authority.

line 249. What does “shows basic affinity to one of these authoritative treatments” mean? Does it or does it not match a given list? Becomes clearer several paragraphs down, but make it clear here.

line 269. What does “associate equal concepts among the authorities” mean?

lines 282-283. Local experts can be wrong.

lines 289-290. I don’t understand what this sentence is trying to convey.

line 300. What is an “aligned taxon concept”?

Do not use the text to describe tables. Instead, make a statement, then reference the table parenthetically.

line 318. Yes, of course it might require delving into the literature. Doesn’t seem like this even needs to me mentioned.

line 324. Orange County, Florida? Note the state given that it’s not a unique county name.

line 336. Again, here is an opportunity to develop targeted outreach to encourage programs to adopt the best scientific taxonomy available.

lines 336-338. If an intent of the manuscript is to build a template for other taxonomic groups, maybe this is relevant, albeit seems rather obvious. But otherwise seems tangential.

line 343. Again I’m puzzled by why “quantify the challenges” is a goal per se.

line 348. Why not go for the realistic case? I still am missing the point.

line 370. It simply may not be possible to address the very problematic groups. In all likelihood, considerable uncertainty will remain regardless. I don’t think that’s necessarily a deal-breaker for integrating monitoring data.

line 390. Data are plural.

line 398. “bi-directionally nonidentical” is not intuitive. Why coin a new term?

lines 438-440. So what are the steps necessary for operation within a program? Is the expectation that users have some type of knowledge that does not require much of an interface? I’m unclear on how one independently could evaluate the tool if it cannot stand alone.

line 442. Use of quotation marks implies that one does not actually mean the thing in quotation marks. Find a way to say what you mean in a straightforward manner.

The discussion seems primarily promotional. I think it would be a good place for addressing the extent to which taxonomic standardization resolves potential analytical biases. For example, to what extent might biases result from differences in sampling frequency, or misidentification of species, or anything else. Unquestionably taxonomic standardization matters, but there are many other sources of uncertainty to acknowledge and weight. Moreover, I would like to see a thoughtful discussion of the rationale for not requesting more compliance with systematics by individual programs, at least if those programs aim to integrate and to inform ecological understanding and conservation. For example, bird monitoring programs annually revise their lists on the basis of AOS updates. Why should butterfly monitoring programs not be encouraged to revise on the basis of Pelham updates?

Table 1. Indicate the standardization of geographic boundaries (presumably standardized before the table was compiled? If not, must be noted). If standardized, what are the boundaries.

Tables 3 and 4 do not seem necessary. Integrate some examples into the text instead.

Table 5 and Figures 2 and 3 are not formatted well. Don’t just cut and paste from an Excel spreadsheet (at least that’s what it looks like to me). Figure 3 isn’t a figure, it’s a table.

Figure 1. What is the source of this representation?

Figure 5 is not intuitive and likely can be deleted.

Reviewer 2 ·

Basic reporting

This study undertakes to resolve an ongoing limitation facing those looking to integrate datasets from different sources. I think that the manuscript introduces the reason why this can be a challenge, and workable way to overcome this.

1. The manuscript would benefit from being tighter and more succinct.
I think that all the information is in here, but there is repetition and a bit jumbled, mainly in the introduction.

I am not a programmer, but I have a background in taxonomy and recording and felt that the information up to line 159 could be condensed. At present, the introduction mentions concept relating to recording and the integration of data, then explains how programmes have dealt with this within the same paragraph. I think that the information would be better split between taxonomy and programming. For example, you could start with a brief overview of how data from citizen science (CS) programmes are used, the challenges of nomenclature, and how this causes problems in data aggregation. This could be followed by how others have tried to resolve this generally (e.g. from line 124), the paucity of integrated models (e.g. from line 101), the specific difficulty & example (e.g. from line 141), before introducing how you tackle these challenges in this study.

2. Further explanation of taxonomic procedures
Given the open and interdisciplinary nature of PeerJ, I believe that a few well-placed sentences regarding taxonomy might help. It’s alluded to in lines 65-70, but I think a few lines on taxonomic processes might help others understand why species names can be different but not ‘wrong’, as well as species names changing over time (especially as splitting/lumping mentioned later on). For example, a short explanation on how species are described through published papers, and that the community are not obliged to use binomial signed in reclassifications and reference an example (e.g. “Description of a new genus for Euptychia hilara (C. Felder & R. Felder, 1867) (Lepidoptera: Nymphalidae: Satyrinae)” Zootaxa 4012 (3): 525–541). Also a note of synonymy, which is mentioned later, but is still quite esoteric.

Additionally
- The abstract should be condensed into one paragraph. The second paragraph contains a lot description that is in the manuscript and not required at this point (e.g. lines 44-51 could be reduced to saying what you did rather than explaining how).
- The first paragraph of the discussion could be absorbed into the introduction.
- Try to avoid colloquial terms. For example line 384 “On the other hand” (perhaps “Conversely”?) and line 574 “can’t afford to”.
- Line 33 – does ‘long legacy of data’ mean historical records?
- Line 73 – mention GBIF for the first time, yet introduce the abbreviation in line 127.
- Line 73 – What do you mean by ‘associated’ with a voucher specimen? Validated?
- Line 75 – by ‘only taxonomic names’ you mean Latin binomials rather than common names, or names rather than individual organisms?
- Line 143 -Does usage refer to which species name is given to an organism, or grammar related difference (e.g. hyphen used between double barrelled names instead of a space)?
- Line 176 – Is there a reference for butterflies being second most monitored group? Is this globally?
- Line 299 – What is meant by ‘species concept’? I’m assuming that you are referring to a taxonomic unit rather than a biological species concept? Is it the same as ‘taxonomic concept’ on line 317?
- Supplemental Article S1 – I couldn’t see any explanation for colours and initials in the spreadsheet. I think that these would be well placed next to the totals so that this could read as a stand alone document.
The discussion nicely explains how this programme has the potential to be adapted in taxa and scale. The figures and tables really are really helpful in visualising the text, and the raw data provides opportunity for further exploration by the reader.

Experimental design

I have no experience of programming, but the method looks like it could be easily replicated or adapted for other taxa or by other citizen science programmes. I’m very pleased that you consulted the program leaders, which I think will add some gravity to the study (lines 284-290). I’ve seen too many programmes designed without consulting the end user, and species identification is rarely straightforward.

I do think that the method would allow a recorder with an interest in programming to give it a go themselves.

Validity of the findings

I applaud the authors for tackling an ongoing issue in the use data from recording schemes. The reasons why a single list of species names is not used universally used can be tricky to explain in an academic paper (I find that it often relates to personalities), as tactfully noted in lines 329-331.

The authors have outlined why this area requires more attention and how integrated data can be widely used in decision making processes.

I think that this will be an iterative process and the programme refined over time, but I think that making this available to the community will stimulate the attention it deserves.

Additional comments

I see that programme managers have been involved, but have you approached any checklist authors? This is only out of interest, as I noticed Dick Vane-Wright’s name on the NABA checklist, and wondered if anyone else had commented on how this might transfer internationally.

Reviewer 3 ·

Basic reporting

a) clearly written
b) intro does not show context as well as it could because it does not describe how scientists resolve taxonomic conflicts (more comments below). The approach in the paper starts to solve the problem they have identified using NA butterfly survey data but it does not help the reader know if the method that is developed will be widely applicable or not. For instance perhaps this was the state of bird taxonomic 100 years ago or it is the current state of dragonfly taxonomy. For other vertebrates there are competing sites that try to keep track of taxonomic advancements and provide a standard. I am not sure about the taxonomy of other groups. Plants taxonomists are slowly working on standards.

c) structure is good
d) figures seem clear
e) raw data has been supplied

Experimental design

a) research is original
b) it is clear the research is filling in a gap but it is not clear if the approach will have wide use. It think the paper is very important in tackling a problem that the taxonomic community has not tried to solve systematically but one that needs to be solved for the rest of biological studies

c) methods are rigorous and generally well explained but I am not clear about how this research relates to ontological research generally and how resolution methods work for other kinds of ontological work. This is a lack of my expertise

Validity of the findings

a) the paper is novelty but I cannot tell about its impact
b) The conclusion section are about providing context for the work and about what they plan to do than about their results

Additional comments

The authors have tackled a difficult and fundamental problem in biology, namely resolving taxonomic name differences across different taxonomic systems. This problem has challenged biologists across the globe with the creation of systems such as ITIS and GBIF, and the desires of biologists to solve taxonomic problems and to do larger scale spatial analyses.

The linking of common and scientific names to taxonomic concepts and the resolution of differences across names and concepts is a problem that taxonomists as a community have not addressed. These authors are trying to address the problem

I see this paper as a case study that systematically resolves differences among local butterfly checklists by using multiple authoritative accounts and working closely with citizen science groups who collect and manage their local data while using different authorities.

The scientists appeared to be are motived to do these project because the survey groups share a common survey method (Pollard Walks) making the data of high value to ecologists because of the structured nature of the method that includes repeated surveys of the same areas and records search effort and contrasts with the more common analyses of presence only data. The first three paragraphs of the introduction devote lots of words to the value of these kind of survey data but this point does not seem central to the point of the paper. My sense of the central point is that it is independent of what one does with the data. Understanding taxonomic differences is a problem whenever one is trying to work with a group of species. Given that taxonomists do not have a system in place to provide a community vision of species concepts and names, the paper is supplying a solution. In that regard I think other scientists also need to resolve local taxonomic lists but may not be motived by doing it for the same reasons the authors are. I suggest saving the first three paragraphs for a paper that analyzes Pollard walk data. I think readers will be more interested in the how’s and the generality of the approach than a specific application. This is not to say that as a case studies that integration of Pollard survey is in any way inappropriate. I think is makes a nice example.

How does the method described here compare with efforts by other North American groups that have butterfly data across the continent such as NatureServe, BugGuide, BAMONA, iNaturalist and NABA? As I recall NatureServe used a human curation system, iNaturalist started with the NatureServe Standard and now lets users curate the taxonomy to some extent. NABA uses a board of experts who periodically review the literature I think.

How much of your method requires human curation and how much can be done by machines? As taxonomy research advances. how are updates made?

The paper has two empirical results.

1) It does a good job of describing and quantifying the differences found across lists (Tables 1 -).
4
It would be important to know if these categories of differences the same as found in other groups. Are there any comparison data around? If not, it would be good to say so.

2) It describes a method to reconcile the differences between the groups.

I am not sure why Ytow et all 2001 (a Ebbe Neilsen prize winner) is not discussed or cited

Ytow, N., Morse, D.R. and Roberts, D.M., 2001. Nomencurator: a nomenclatural history model to handle multiple taxonomic views. Biological journal of the Linnean Society, 73(1), pp.81-98.

Lines 347-349: this was disappointing to read because I agree that 3rd party resolution is a most important case, at least for synthesis projects. Could you take a stab at this so people could than make use of the butterfly data for other analyses?

Line 438: Tool is not meant to stand alone Glad to know this but it means there is not a tool for other use


I think the paper has value because it systematically identifies taxonomic issues about differences among authorities. This can channel actions that need to be resolved by the taxonomists. This value is not explicitly stated.

---

## Round 0.2 · Minor Revisions

All reviewers appreciated the effort you put in the revision, and I agree with their assessment. There are now only minor concerns left that you should be able to address quickly.

Reviewer 1 ·

Basic reporting

see below

Experimental design

see below

Validity of the findings

see below

Additional comments

This manuscript is much improved. However, in many places the writing can be clarified or streamlined, and in a few instances somewhat defensive language could be toned down. Also, it’s fine to briefly mention a planned extension of work presented in a manuscript, but extensive description of plans (e.g., lines 875-898) is not convincing. All of us would like to do a lot of things that may or may not happen, sometimes for reasons beyond our control.

line 32. Change “interpretations” to “identifications” or “names”

line 33. “coercing” is a rather loaded word. Change to “encouraging”, and, to avoid repeating the same word, remove the “encouraging” after “While”.

line 39. Again, “interpretation” doesn’t seem accurate

line 41. “curate the term” is not clear

line 45. “scaffolding” is unclear

line 254. What’s a “truly” integrated analysis?

line 257. Change “between” to “among”

line 258. This might be a trend in citizen science, but not in data collection by trained lepidopterists.

line 263. And with names only, especially when data collectors are not experts, it’s less certain whether the identifications are correct.

line 292. The relationship between the concept and the taxa? Not clear.

line 294. Not really reinterprets, although there are reclassifications on the basis of morphology, genetics, life history, and other types of data.

line 300. Why is unstable in quotation marks? Do you not actually mean unstable? Same with line 304 (what do you actually mean?) and elsewhere (line 357, 545, 554, 698, 752, 792)

line 307. What do you mean by “taxonomic interpretation of the data source”?

line 314. What do you mean by reanalysis of names? “inscribe” is not the correct word.

line 315. Remove “event”

line 328. What is a “system of web-services”?

line 399. Change “demonstrating” to “testing”

line 411. “thus far unanimously agreed”—this is vague. Which programs have agreed?

line 462. Clearer to say “at the genus or family level”

line 465. As opposed to inappropriately combined? Sufficient to simply say “can be combined”

line 470. Should be evaluating, not demonstrating.

line 514. Taxonomic, not biodiversity. Biodiversity is a far more comprehensive term; it is not synonymous with species or other levels of orgainzation.

line 521. Either explain “official” or delete.

line 524. “Opler and Warren” is short. Throughout the manuscript, spell out rather than abbreviating. Same for species (line 600 and elsewhere), receiver (line 642 and elsewhere), and donor (line 643 and elsewhere).

line 549. Change “must” to “would”

line 550. “personal” not “personnel”. How many is “all”?

line 551. Delete “Conveniently”

line 679. Delete “exactly”

line 698. bi-directionally nonidentical really is a mouthful and not easily understood. Please work to find a clearer term.

line 893. Not clear. Do you mean different taxonomic levels?

line 910. Delete “100% of”

line 928. Conclusions are unnecessary. Delete any material covered above, and integrate any novel material into the discussion.

Reviewer 2 ·

Basic reporting

The manuscript has been much improved and reads in an easy to follow manner

Experimental design

Integrating such data is as much about the data as it is the data providers, and this is handled sensitively in trying to achieve the main outcomes. Even if all parties agreed, taxonomic classification is dynamic, and this has been accounted for.

Validity of the findings

The program is very targeted, but consequently a great opportunity to trial within a specific taxon group and geographic area. I imagine that tweaks will be required once the programme is widely used, and I get the impression that the authors would welcome the opportunity to increase usability where required.

Additional comments

I found the revised manuscript easier to follow and understand. Good luck with everything!

Reviewer 3 ·

Basic reporting

The revisions improve the manuscript. The authors had good responses to the reviewers questions and comments

Experimental design

OK

Validity of the findings

It is an old problem in biodiversity informatics. Different solutions have been proposed. This is a novel one

Additional comments

The revision improve the manuscript. I think you illustrate a method that can be developed into a tool.

Some comments

1) Title
Is it a system or a method?


2) Abstract

First sentence addition “Integrative modeling methods, such as niche modeling,”

Last sentence to add? “After integration, these taxonomically standardize datasets can subsequently be used as input for many kinds of biodiversity analyses.  


3) line 346 This was well before the current global explosion of citizen science monitoring programs ( see Pocock et al 2015 for development in Great Britain) ,

Pocock, M.J., Roy, H.E., Preston, C.D. and Roy, D.B., 2015. The Biological Records Centre: a pioneer of citizen science. Biological Journal of the Linnean Society, 115(3), pp.475-493)

4) line 501-506 can delete the word “Objective”

5) line 601 Comment For phylogenetic comparisons, these issues would be good to resolve.

---

## Round 0.3 · accepted · Accept

Thanks for carefully revising the paper at each step. Your paper addresses an important issue (and not just for citizen science - in my own work I have been confronted with such taxonomic issues and they are often frustrating but necessary to address). I do hope your approach will be used.